# A Sustainable Career Perspective of Work Ability: The Importance of Resources across the Lifespan

**DOI:** 10.3390/ijerph16142572

**Published:** 2019-07-18

**Authors:** David Stuer, Ans De Vos, Beatrice I.J.M. Van der Heijden, Jos Akkermans

**Affiliations:** 1Antwerp Management School, 2000 Antwerp, Belgium; 2Faculty of Business and Economics, University of Antwerp, 2000 Antwerp, Belgium; 3Institute for Management Research, Radboud University, 6525AJ Nijmegen, The Netherlands; 4School of Management, Open University of the Netherlands, 6419AT Heerlen, The Netherlands; 5Faculty of Economics and Business Administration, Ghent University, 9000 Ghent, Belgium; 6Hubei Business School, Hubei University, Wuhan 368 Youyi Ave., Wuchang District, Wuhan 430062, China; 7Kingston Business School, Kingston University, London KT11LQ, UK; 8School of Business and Economics, Vrije Universiteit Amsterdam, 1081HV Amsterdam, The Netherlands

**Keywords:** perceived work ability, meaningfulness of work, perceived fit with current job, future-orientedness of the job, sustainable careers, age

## Abstract

In this study, we examine employees’ perceptions of their work ability from a sustainable career perspective. Specifically, we investigate the role of a person’s perceived current fit (i.e., autonomy, strengths use and needs-supply fit), and future fit with their job as resources that affect perceived work ability, defined as the extent to which employees feel capable of continuing their current work over a longer time period. In addition, we test whether meaningfulness of one’s work mediates this relationship, and we address the moderating role of age. Our hypotheses were tested using a sample of 5205 employees working in diverse sectors in Belgium. The results of multi-group Structural Equation Modelling (SEM) provide mixed evidence for our hypotheses. While all four resources were significantly and positively related to perceived meaningfulness, only needs-supply fit was positively related to perceived work ability. Strengths use, on the other hand, was also significantly related to perceived work ability, yet in a negative way. These findings underscore the importance of distinguishing between several types of resources to understand their impact upon perceived work ability. Interestingly, the relationship between future-orientedness of the job and perceived work ability was moderated by age, with the relationship only being significant and positive for middle-aged and senior workers. This suggests an increasingly important role of having a perspective of future fit with one’s job as employees grow older. Contrary to our expectations, meaningfulness did not mediate the relationships between resources and perceived work ability. We discuss these findings and their implications from the perspective of sustainable career development.

## 1. Introduction

Work takes up roughly one third of the day for a large portion of the adult population, and this continues for a very large share of one’s life. The influence of work is pervasive in many domains of life, and has important consequences for one’s life satisfaction [1], health [2], and subjective career success [3], to mention but a few. Current labor market trends such as increasing automization and robotization of work [4], and organizational contexts characterized by volatility, uncertainty, complexity, and ambiguity [5] challenge the extent to which employees experience a strong fit between their work-related needs and what their work offers them. These labor market trends may have a considerable impact on the sustainability of people’s careers if individuals struggle to achieve such a strong fit for extended periods of time [6,7]. Indeed, the ageing of the working population provides economic pressures to motivate citizens to work longer, which makes the question of sustainable careers across the life-span even more important [8]. Therefore, this study examines work ability from a sustainable career perspective.

In particular, in this paper we study *perceived* work ability, referring to it as a worker’s general feelings or perceptions regarding their capability to continue doing their current work towards the future. Work ability is a holistic concept that refers to people’s ability to do their work in a healthy and productive way given the balance between a person’s resources—including their health and functional abilities, education and competence, and values and attitudes—and their work demands [9,10,11]. As such, at its core, ‘perceived work ability’ refers to employees’ perceived ability to work. We approach perceived work ability from a sustainable career perspective, thereby considering it as an indicator of a sustainable career [7]. Indeed, in their conceptual model on sustainable careers, De Vos et al. [6], argue that (occupational) health is one of the core indicators of career sustainability, thereby referring to healthy, happy, and productive workers [12]. Also, work ability—and more generally: health and well-being—has been a core topic of career research in recent years [13], further emphasizing the importance of studying work ability as part of career sustainability. Thus, although work ability in itself does not comprise a sustainable career, we argue that it does provide an indication of one’s career sustainability.

More specifically, we focus on the resources that can enhance a person’s perceived work ability. In particular, we first examine the association between a person’s current fit with their job (in terms of autonomy, strengths use and needs-supply fit) and their perceived work ability. Second, we also incorporate a person’s perception of future fit (i.e., future-orientedness) with their job. We postulate that these specific resources of current and future fit are especially important to examine as antecedents of perceived work ability given the rapid changes in today’s world of work. In the current labor market, jobs that are a good fit in the here and now, and that also remain so across the life-span are key to career sustainability.

Furthermore, with a growing emphasis on the idiosyncratic nature of careers, one of the core dimensions of a sustainable career is finding and retaining work that provides meaning to the person [7]. Yet, having work that brings meaning to people’s lives is becoming an ever-increasing struggle due to the above-mentioned societal challenges. We argue that experiencing meaningfulness in one’s work is important for career sustainability [6] and we empirically explore the role of perceived meaningfulness as a mediator in the relationship between resources (i.e., current and future fit of the job) and perceived work ability.

Thus, departing from a sustainable career perspective, the first contribution of this paper is to study the role of resources in people’s perceived work ability, through the mediating role of perceived meaningfulness. In doing so, rather than considering work from the demand side, we focus on the potential resources that work may bring to a person, and how these resources add to self-perceived work ability across the life-span via meaningfulness of one’s job. As a second contribution, by examining both current and future fit of one’s job and its role in perceived work ability, the dimension of time (see [6], for the process model of sustainable careers) is added into our research framework. Third, given the fact that events and evolutions in the person and their context impact their experiences, and may bring along different needs, challenges, problems, and opportunities [6,14] we hypothesize that the proposed relationships may differ depending on one’s career stage. Therefore, we test whether age moderates these relationships by considering three different age groups: young workers (20–34 years), mid-career workers (35–49 years), and senior workers (50+) (cf. [15]). In doing so, we add to the existing work ability and sustainable careers literature by providing further empirical insight into the influence of antecedents, the possibly mediating role of perceived meaningfulness, and the role of age as a possible moderator in understanding what factors explain perceptions of work ability. Figure 1 depicts our research model.

### 1.1. Perceived Work Ability from a Sustainable Career Perspective

The idea that careers reflect the continued employment of individuals in jobs that facilitate their personal development over time has been the underlying ideology of career research for a long time [16]. Analogously, the notion of sustainable careers approaches the career from a dynamic and systemic perspective, arguing that multiple stakeholders play a key role, such as one’s family, peers, supervisor, and employer [6]. As such, the sustainable career paradigm considers how a person can foster person-career fit over time by generating new resources through one’s work rather than depleting them [6,7]. Health, happiness and productivity [12] are considered as the three indicators of a sustainable career. Health encompasses both physical and mental health, and refers to the dynamic fit of the career with one’s mental and physical capacities. Happiness concerns the dynamic fit of the career with one’s values, career goals, and needs. Productivity means strong performance in one’s current job as well as a guaranteeing a high employability or career potential towards the future [17].

Seen from a sustainable career perspective, perceived work ability refers to the individual’s general feelings or perceptions to continue performing their current work. More specifically, it expresses how well the individual resources meet the requirements of the job ([18], p. 393). Perceived work ability implies the anticipated experience of balance between personal resources and work demands across the career, and has been found to be a strong predictor of early retirement from the labor market [18,19,20]. We argue that perceived work ability is an important element of a person’s long-term health, and thereby forms an indication of someone’s career sustainability [6].

To achieve high levels of work ability, it is important that an employee is mindful about what matters to them [12], and creates opportunities for a meaningful existence [21,22]. As such, perceived work ability has two important components: a developmental and an individual component [6]. First, the developmental component underlines that the employee builds upon and expands their resources over a longer period of time, preferably across the entire career. This puts an emphasis on how well one’s work protects and enhances one’s current as well as one’s future perceptions of work ability, thus incorporating the capacity to flourish both in the here-and-now and in the future.

Second, the individual component of perceived work ability underlines that work should lead to a personally meaningful existence, and should be understood from a fundamentally individualistic perspective [23], as careers form a complex mosaic of objective experiences and subjective evaluations [6]. This complexity has a strong impact on the meaningfulness of one’s work, and explains its highly idiosyncratic character (ibid.).

Consistent with these arguments, in this study we focus on the association between resources and perceived work ability, thereby incorporating not only a person’s perceptions of one’s current but also of one’s future fit with the job. Moreover, in this relationship, we expect perceived meaningfulness to play a critical role as a mediator, and we posit that age might moderate the pattern of relationships.

### 1.2. Resources and Perceived Work Ability

Research on antecedents of work ability is quite extensive, though most studies thus far have focused on work-related demands that might undermine work ability (e.g., [24,25,26]). Yet, some recent studies show that next to demands, work-related resources are also critical for one’s work ability. For example, Airila et al. [27] showed that task resources in the form of autonomy and strengths use were predictive of a higher work ability. In addition, Airilia, et al. [28] showed that this positive effect of task resources even holds over a longer time period. In a similar vein, Pohjonen [29] also showed a positive relationship between autonomy and work ability. Thus, prior studies have shown that both autonomy and strengths use in one’s job are important resources for enhancing one’s work ability.

In this study, we replicate these findings and extend them by introducing the notion of fit as an important resource in relation to perceived work ability. Person-job fit and—seen from a sustainable career perspective, person-career fit—are crucial resources that can lay the foundation for career sustainability [6]. Departing from the notion of a person’s perceived fit with their job, perceived work ability can be understood by looking at the extent to which a person experiences a fit between their specific needs and what the job actually provides. This line of argumentation corresponds with prior findings on the importance of *autonomy*, *strengths use* and *needs-supply fit*, as these resources allow individuals to establish a fit between their competencies and the work they do [30], for how career competencies and job crafting relate to each other). To further explain, first, autonomy refers to the degree to which a job allows freedom and discretion to schedule one’s work and make decisions about it [31]. It is a key resource for individuals across the lifespan to enhance their long-term well-being and performance (e.g., [32,33,34]) and, thus, their perceived work ability. Second, as summarized by Kong and Ho [35], an individual’s strengths use at work refers to traits or capacities that are nurtured with increasing knowledge and skills [36,37]. In addition, Govindji and Linley [38] note that strengths use enables authentic expression, and that it energizes people. Third, a key resource in light of current fit with a job is that it should fit well with one’s personal needs and values, which is captured in the notion of needs-supply fit [39]. In all, we propose that autonomy, strengths use, and needs-supply fit are key resources that represent current fit with one’s job, and that they are important resources for achieving work ability.

By approaching perceived work ability from a sustainable career perspective, we postulate that also an individual’s anticipation of future fit with their job will operate as a resource that might enhance work ability. More specifically, we define future-orientedness of one’s job as the perceived availability of long-term fit between the person’s needs and competencies, and what the job offers them, and consider this an antecedent of perceived work ability. Preparedness for future events in one’s personal life is important in guiding the individual towards positive future outcomes [40]. In a similar vein, jobs need to provide beneficial opportunities for the future in order to be considered sustainable for its holders [22]. Not addressing one’s future needs at the workplace might result in long-term misfit, because jobs tend to evolve over time, due to all kinds of environmental and labor market changes. Moreover, working organizations themselves change as well [41]. As the work context becomes increasingly volatile and jobs change or disappear, it follows that skills that are relevant in today’s labor market might not stay relevant in the longer run, herewith underscoring the importance of looking at both current and anticipated future fit with one’s job as antecedents of perceived work ability.

To summarize, we argue that perceived current and future fit with one’s job are critical resources that are advantageous in the light of individuals’ perceived work ability. Therefore, we formulate the following hypothesis:

**Hypothesis** **1:**
*Resources in the form of (a) autonomy, (b) strengths use, (c) needs-supply fit, and (d) future-orientedness of one’s job will be positively associated with perceived work ability.*


### 1.3. The Mediating Role of Perceived Meaningfulness of Work

Research has clearly shown that performing meaningful work provides richer, more satisfying and more productive employment for individuals [42]. Meaningfulness of work refers to the extent to which an individual employee derives positive meaning from work [2] and results from the match between work and different domains of the self (i.e., values, beliefs, and norms) [23]. This implies that meaningfulness of work is closely related to the concept of self and is central to one’s personal identity, as it articulates the role of specific values, beliefs, and norms in the perception of meaningfulness of work. Hence, perceived meaningfulness of work is an important aspect of personal well-being [43]. Building further on this line of thinking, work becomes meaningful because it provides the opportunity to realize an idealized self [43] and to satisfy one’s personal needs [44]. Based upon this line of reasoning, and applying a resource management perspective [45], we propose that the resources included in our model—that is, perceived current fit and future fit with one’s job—will positively relate to meaningfulness of work.

In turn, we expect meaningfulness of work to be related to perceived work ability. Departing from a sustainable careers perspective [6], we posit that meaningfulness of work is an important factor in explaining how employees assess their long-term work ability. In particular, when people experience current and future fit with their jobs, this allows them to realize an idealized self through work and create opportunities for meaningful existence [43]. In turn, this will positively affect their perception of the extent to which they feel capable to continue doing their current job, that is: their work ability. In all, we hypothesize that—in addition to the direct relationship between resources and perceived work ability, as formulated in Hypothesis 1—the resources of current and future fit with one’s job are likely to enhance perceived work ability via meaningfulness of work. Stated differently, when individuals consider their work to provide them with autonomy, a high level of strengths use and need-supply fit, and also provide a good perspective for future fit, this will generate a sense of meaningfulness, which will then enhance their perceived ability to continue doing their job over a longer time period.

**Hypothesis** **2:**
*Resources in the form of (a) strengths use, (b) autonomy, (c) needs-supply fit, and (d) future-orientedness of one’s current job will be positively associated with perceived meaningfulness of work.*


**Hypothesis** **3:**
*Perceived meaningfulness of work will be positively associated with perceived work ability.*


**Hypothesis** **4:**
*Perceived meaningfulness of work will partially mediate the relationship between resources and perceived work ability.*


### 1.4. Resources, Meaningfulness of Work, and Perceived Work Ability across Age Groups

Building upon the notion that values, beliefs, and norms are dynamic throughout the life-span [46], we posit that individuals prioritize things differently throughout their career. This implies that the hypothesized relationships in our research model may differ for people being in different career stages (see also [6]). This makes work ability a somewhat elusive concept, because we assume it to be dynamic throughout the lifespan [7]. Following from this line of reasoning, what motivates people in the beginning of their career may vary from what motivates them in the midlife career stage, and/or at the end of their career, since perspectives on time, mortality, and the developmental tasks that are inherent to different career stages also change [47]. Therefore, we differentiate between three groups of workers based upon their career stage, i.e., young workers, mid-career workers, and senior workers [15]. This division categorizes workers into groups with a similar range, thereby considering a separate category for the middle-aged employees (aged 35–49 years), which roughly corresponds to the ‘mid-career’ category (see also [48,49].

Given the observation that personal needs tend to evolve and change in terms of their relative importance throughout the career [6,14], an important question becomes to what extent employees’ perceptions regarding their perceived work ability are driven by the importance of different foci in life, depending on their career stage. According to the life-span theory of Selection, Optimization and Compensation (SOC) [50], the selection of relevant life goals that are aligned with one’s important foci over time is a developmental task that becomes more important as we age [46]. SOC theory further makes a conceptual difference between two types of selection in order to maximize gains and to minimize losses that individuals experience over time: elective selection and loss-based selection, respectively. The former is a selection of goals that are driven by a match between an individual’s needs, while the latter selection of goals is based on a loss of resources. In order to maximize gains, individuals select outcomes or goals that are desirable (i.e., elective selection), and optimize their resources (cf. COR theory [51]) to reach these. To minimize losses, individuals select fewer goals in response to (foreseen) losses, and compensate for these losses by investing their remaining resources in counteracting these losses (cf. primacy of resource loss).

SOC theory predicts that the allocation of resources aimed at growth will decrease with age, whereas the allocation of resources aimed at maintenance and regulation of loss prevention will increase with age [50]. Correspondingly, Freund [52] found a shift in regulatory focus from being aimed at promotion for younger individuals to focusing on maintenance and prevention in later life. In the context of our study, we argue that meaningfulness in one’s job will be more important for older employees as they are relatively more focused on maintaining what they currently have and preventing losses, compared with younger employees who are more focused on striving for future opportunities and growth in one’s current or in other jobs. Analogously, following Socio-emotional Selectivity Theory (SST) [53,54], which states that people prioritize meaningfulness of interactions because their future time perspective is starting to get limited [54], we argue that, with ageing, resources that strengthen the meaningfulness of work gain in importance. After all, when growing older, in general, people shift their motive for social interaction, in our case at the workplace, from gaining resources, such as money and/or promotion (i.e., instrumental) towards receiving affective rewards (i.e., emotional). In sum, adopting a sustainable career perspective and following SOC theory and SST, we assume that the resources of current and future fit with one’s job gain importance across career stages as antecedents of meaningfulness of work and perceived work ability. This leads to our final study hypothesis:

**Hypothesis** **5:**
*Age will moderate the mediated relationship between resources and perceived work ability via meaningfulness of work, such that the relationship is strongest for senior workers compared to, respectively, mid-career and young workers.*


## 2. Methods

### 2.1. Sample and Procedure

Data were collected in collaboration with a leading newspaper in Flanders which is the Dutch speaking region in Belgium. They distributed a link to the online survey via their online and printed communication channels. Participation in the survey was entirely voluntary. Respondents received the results of their survey after they filled in the questionnaire such that they obtained a personal profile based upon their score on each of the core variables measured. Data was scrubbed of identifying information. The dataset used in the analysis contained 5205 responses after excluding participants with missing data. 44.1% of the sample are men, 55.9% women. Mean age was 39.52 years (SD = 10.199) and respondents had changed functions on average 1.56 (SD = 1.950) times up until now in their careers. Furthermore, they had been working on average for 11 years (SD = 9.320). Age categories were defined in accordance to different career stages, with the younger category being those between 20 and 34 years (*n* = 1959), mid-career workers in the age category between 35 and 49 years (*n* = 2270), and senior workers, being 50+ (*n* = 976) [14]. One could argue that our hypotheses are linear, in the sense that we assume linearity in the strength of the interaction across age. Therefore, it would be logical to keep age as a continuous variable to test for interactions. However, we opted to categorize age, since this approach has the advantage to model possible non-linearities in relationships and is roughly consistent with early, mid and the late career stages [55].

### 2.2. Measures

All scales were measured using 5-point Likert scales ranging from 1 (*completely disagree*) to 5 (*completely agree*).

*Autonomy* was measured with a 4-item scale from the VBBA [56]. Cronbach’s alpha was 0.77. An example item was: “I have a lot of autonomy in how I do my job”.

*Strengths use* was measured with a 3-item scale based on Kong and Ho [35]. Cronbach’s alpha was 0.82. An example item was: “My work allows me to apply my talents”.

*Needs-supply fit* was measured with the 3-item scale of Cable and DeRue [39]. Cronbach’s alpha was 0.89. An example item was: “My work offers me everything that I search for in a job”.

*Future-orientedness of the job* was measured with five items based on the future time perspective scale from Strauss and colleagues [57] which we reformulated to represent the perspective of future-orientedness offered by the job itself. Cronbach’s alpha was 0.82. Example items are: “I expect to do many interesting things in my job in the future” and “In my current job I develop competencies that will keep me employable in the future”.

*Meaningfulness of work* was measured with two items from the positive meaning scale [58]. Cronbach’s alpha for this scale was 0.782. An example item was: “I consider my work to be meaningful”.

To capture *Perceived work ability*, we assessed the degree to which respondents felt capable of continuing doing their current work using three newly developed items: “I don’t see myself continuing to work in my current job for much longer” (reversed scoring), “I feel able to continue working in my current job until I retire” and “A higher retirement age is not a problem for me personally”. Cronbach’s alpha was 0.61.

### 2.3. Analytical Strategy

We employed structural equation modelling to test our conceptual model and used the lavaan package (0.6–3) in R 3.5.2 to analyze the results [59]. To test whether a model was a good description of the data, we used a combination of fit indices, as is advisable when performing structural equation modelling [60,61]. We used the following cut-offs: CFI > 0.95, TLI > 0.95 and RMSEA < 0.05 for good fit and CFI > 0.90, TLI > 0.90, RMSEA < 0.08 for adequate fit [60]. First we constructed a general measurement model, then used multiple group confirmatory factor analysis (MG-CFA) to see if the measurement model is invariant across the different age categories. If we could establish at least partial metric invariance, we went to the second step, which involved testing our theoretical model. Metric invariance was established by comparing the change in global fit indices. If there was a drop in either of the global fit indicators, we would look if items have the same loading across age categories. We do not use a Chi-square-test, since this might lead to an oversensitive test given the size of our sample. In the second step we compared multiple alternative models to our theoretical model to see whether these comparisons would support our theoretical model. Our theoretical would be seen as better if it shows the best fit to the data. Thirdly, we tested for invariance in the structural model, by constraining parameters one by one at the structural level across different age categories as a test for age interactions and as such for examining moderated mediation. If constraining a parameter led to substantial misfit, as indicated by the Chi-square test, the parameter was assumed to be different across age categories and set free across age categories.

## 3. Results

### 3.1. Measurement Model: CFA across Age Groups

First, a general measurement model was constructed for the total sample and this was compared to a single factor model to test for common-source bias [62]. Fit was inadequate for the single factor model (Chi-square (170) = 8466.270, CFI = 0.854, TLI = 0.837, RMSEA = 0.097), meaning that common-source bias is an unlikely explanation for the relations found in the study. Initial model fit was adequate (Chi-square (155) = 3363.431, CFI = 0.943, TLI = 0.931, RMSEA = 0.063). Using a combination of modification indices and theoretical reasoning, we covaried three pairs of items. The first pair was: ‘I am encouraged to develop new skills in my job’ and ‘I gain experience at work in a variety of domains where I can broaden my knowledge and skills’ which pertains more to a developmental side of future-orientedness. The second pair was in the autonomy scale: ‘I have a lot of autonomy to decide how I do my work’ and ‘I have influence over my department’s decisions’, both referring more to the personal power expressed in autonomy in comparison with the other items (e.g., ‘I decide with others how the tasks are distributed (‘who does what?)’). The last pair of covaried items was also in the future-orientedness scale: ‘As far as my work is concerned, I still see many opportunities for myself in the future’ and ‘I expect that in the future I will be able to do many exciting new things in my work’. The logic for this last pair is that these items both make a direct reference to the future, thus providing more common ground than the other items. This led to a substantial increase in model fit, leading to good model fit (Chi-square (152) = 2365.332, CFI = 0.961, TLI = 0.952, RMSEA = 0.051).

At this level we also extracted the correlation matrix (Table 1). Looking at the matrix, there was a very high correlation (*r* = 0.934) between strengths use and needs-supply fit. To test whether these two variables might be reduced to one factor, a model was tested wherein both factors were merged together. However, its global fit was substantially worse than the previous model (Chi-square (160) = 3774.537, CFI = 0.936, TLI = 0.924, RMSEA = 0.066, Δ CFI = −0.025, Δ TLI= −0.028, Δ RMSEA = 0.015). This is an indicator that concatenating these factors is inappropriate, leading us to keep both as separate factors.

In order to test for the age interaction at the structural level, we first needed to test whether the same factor model held across different age groups and, subsequently, at least partial metric invariance needed to be established [61]. As such, we tested for configural equivalence. The fit for the configural model was generally satisfactory, indicating that the same factor structure could be preserved across different age groups (Chi-square (456) = 2680.907, CFI = 0.961, TLI = 0.952, RMSEA = 0.053). Next, metric equivalence was tested in the factor model. We did this by comparing the fit indices between the configural and the metric model. When constraining the loadings there was a very small difference between the configural and metric model in CFI (ΔCFI = 0.001). By using modification indices, we released equality of loading constraint for one item: “I don’t see myself continuing to work in my current job for much longer”. This item had a higher loading in both the middle and older age categories, indicating a greater importance for this item when measuring the construct in these groups of employees. This possibly reflects a greater proclivity towards thoughts of retirement. As such, this might help explain our lower reliability for the perceived work ability scale, since a lower loading in one of the distinguished age categories can be associated with a lower Cronbach’s alpha. Final fit for the model was practically the same as for the configural model (Chi-square (456) = 2751.340, CFI = 0.961, TLI = 0.953, RMSEA = 0.052), meaning that partial metric invariance was tenable as an assumption. As such, we could proceed to investigate whether there were structural differences in the model across age categories.

### 3.2. Structural Model

Before testing the model across the three different age categories, we first tested the overall structure of our theoretical model in the total sample. We also compared a series of plausible alternatives to our hypothesized model. This is considered good practice in SEM and will strengthen our belief that our current model is suitable [61]. As not every model was nested in the other, we could not use Chi-square tests to compare them. Instead, we compared the AIC indices of different models since this index is suited for comparing non-nested models [61]. The results can be found in Table 2. This procedure led us to conclude that the model that included direct paths (Model D) had the best fit to the data and, as such, this model was retained. This model also allows us to test for partial mediation, which will be discussed below.

We tested for differences in the structural models between the three age groups by constraining regression parameters to be equal across age categories one by one. Since the models are nested versions of one another, it is appropriate to use Chi-squared tests in these instances [61]. If placing constraints led to a substantial misfit in a subsequent model, the parameter was set free. Nine individual hypotheses were tested, increasing the chance of spurious findings and this is the reason for applying a Bonferroni correction to the alpha value of the tests [63].

Accordingly, in order to be deemed a significant misfit, the *p*-value needed to be below 0.0056. This led to nine models that were tested. The final model retained was Model 9 in Table 3, which allowed for an interaction of age on the relationship between future-orientedness of the job and perceived work ability. Fit of the final model was good (Chi-square (498) = 2777.124, CFI = 0.960, TLI = 0.955, RMSEA = 0.051). The results of the final model are displayed in Table 4. First, regarding the relationship between resources and perceived work ability (*Hypothesis 1*), we only found a significant and positive association between needs-supply fit and perceived work ability (*β* = 0.777, *p* < 0.001 for the three age categories) and between future-orientedness of the job and perceived work ability (young: *β* = 0.196, *p* < 0.001; middle-aged: *β* = 0.275, *p* < 0.001; senior: *β* = 0.272, *p* < 0.001). Contrary to our expectations, a significant negative relationship was found between strengths use and perceived work ability (*β* = −0.298, *p* < 0.001 for the three age categories). Finally, the relationship between autonomy and perceived work ability was non-significant (*β* = −0.017, *p* = 480). Together, these findings provided mixed support for *Hypothesis 1*.

In general, the associations between the four resources included in our model and meaningfulness of work were in line with *Hypothesis 2*. Firstly, autonomy had a significant positive relation to meaningfulness of work (*β* = 0.127, *p* < 0.001 for the three age categories). Secondly, we found a significant positive association between needs-supply fit and meaningfulness of work (*β* = 0.351, *p* < 0.001 for the three age categories). Thirdly, there was a positive relationship between strengths use and meaningfulness of work (*β* = 0.321, *p* < 0.001) for the three age categories). Future-orientedness of the job was also significantly and positively related to meaningfulness of work (resp. *β* = 0.144, *p* < 0.001 for the three age categories).

Contrary to our expectations (*Hypothesis 3*), the relationship between meaningfulness of work and perceived work ability was not significant (*β* = 0.039, *p* = 0.270). As there was no significant statistical relationship between meaningfulness of work and perceived work ability, we could not further test for mediation. As such *Hypothesis 4*, which was our mediation hypothesis, was not supported by our data.

The results of our multi-group analysis provided limited support for *Hypothesis 5*. Age only appeared to moderate the direct relationship between future-orientedness of the job and perceived work ability, such that the relationships were significantly weaker for younger workers (*β* = 0.196, *p* < 0.001) than for their middle-aged and older counterparts (*β* = 0.275, *p* < 0.001; *β* = 0.272, *p* < 0.001, respectively). The difference between these three parameters is significant, given that Model 8 entailed a significant misfit compared to its previous iteration (Chi-square (2) = 11.0278, p = 0.004), thus causing us to free these parameters.

The other relationships in our model did not differ depending on the respondents’ age category. These last results, in combination with the finding that meaningful work, our mediator, was not significantly related to perceived work ability, suggest that we could not find support for moderated mediation. Accordingly, in order to be deemed a significant misfit, the *p*-value needed to be below 0.0056. This led to nine models that were tested. The final model retained was Model 9 in Table 3, which allowed for an interaction of age on the relationship between future-orientedness of the job and perceived work ability. Fit of the final model was good (Chi-square (498) = 2777.124, CFI = 0.960, TLI = 0.955, RMSEA = 0.051). The results of the final model are displayed in Table 4 and in Figure 2.

## 4. Discussion

In this paper we adopted a sustainable career perspective to examine the role of resources (i.e., current and future fit with one’s job) as antecedents of perceived work ability, and the mediating role of meaningfulness of work. In addition, using moderated mediation modelling we tested whether this model would hold for workers from three different career stages (i.e., young workers, mid-career workers, and senior workers). Our hypotheses were tested using a large sample of Belgian workers. The results provide mixed evidence for our hypotheses. While all four resources were significantly and positively related to perceived meaningfulness, only needs-supply fit was positively related to perceived work ability. Strengths use, on the other hand, was also significantly related to perceived work ability, yet in a negative way.

### 4.1. Theoretical Contributions

A first theoretical contribution of this study is our focus upon resources as antecedents of perceived work ability. Most of the research to date has predominantly focused on job demands that negatively affect a person’s perceived work ability such as physical workload, conflicts at work, and stress [19,20]. Bringing in a resource perspective is an important addition to existing literature because resources can buffer demands and also have a unique motivating potential themselves [28]. Specifically, we expected that both the perceived current and future fit of the job would be positively related to perceived work ability via meaningfulness of work. Our results were mixed. Of the resources in this study, only one indicator of current fit—needs-supply fit—related positively to perceived work ability. Thus, if the job fulfills the person’s psychological needs and preferences [61], this is likely to enhance one’s ability to continue doing their job now and in the longer term. However, contrary to our expectations, this relationship was not mediated by meaningfulness of one’s work, even though needs-supply fit did relate significantly to meaningfulness of work. These findings suggest that the fulfilment of psychological needs is a direct predictor of meaningfulness of one’s work and perceived work ability, rather than the expected indirect relationship in which meaningfulness of one’s work would mediate between needs-supply fit and perceived work ability.

Surprisingly, we neither found a significant association between autonomy—a second indicator of current fit with the job—and perceived work ability, nor a mediated relationship via meaningfulness of work. Apparently, although autonomy is a key resource in enhancing well-being and performance [28], it does not relate directly to one’s perceptions of work ability. Yet, our findings do suggest that work is felt as more meaningful as autonomy increases. Whilst the latter is in line with earlier findings, the former is in contrast to an abundance of literature stressing the importance of autonomy in work-related outcomes, such as the Karasek model [62], and Self-Determination Theory [63]. One possible explanation for our outcome is that more autonomy in one’s work might also bring along additional challenges, herewith aggravating the burden in terms of self-management when autonomy is accompanied by a stronger focus on results and high performance goals. As shown in scholarly work on the influence of New Ways to Work, it is important that employees *experience* their working conditions as resources instead of demands in order to result in positive outcomes [64]. Future work using more specific measures of different forms of autonomy might shed more light on this issue.

The third indicator of current fit with one’s job, strengths use, also showed surprising results. Although it related positively to meaningfulness of work as hypothesized, contrary to our expectations, it related negatively to perceived work ability. Initially, we were surprised with this finding given prior evidence for strengths use as a predictor of well-being [65] However, there is some empirical evidence suggesting that in order for work to be deeply meaningful one also needs to ‘suffer’ for their craft [66]. In line with this argumentation, prior studies found that challenging job demands can work both as a motivator and a stressor [67]. Consequently, one can speculate that strengths use, besides being an attractive resource, may also instill these potentially harmful aspects and inspire people to work ‘to the bone’. Thus, even though a high level of strengths use is likely to provide a sense of meaningfulness in one’s work, it can also have a potential dark side of undermining work ability when people are too highly involved in their job. Furthermore, the sustainable career paradigm may also provide a further explanation. From this perspective, personal investment in one’s current job might lead to depletion of resources thereby lowering the sustainability of one’s career over time [6]. Reduced health – operationalized in this study in terms of perceived work ability—might be an important indicator of this phenomenon. A related explanation can be found in research on workaholism, from which we know that the mechanism of controlled motivation might explain the negative impact of ‘working too hard’ on employee outcomes, compared with the positive impact of engagement [68,69]. The research on strengths use, stemming from the domain of positive psychology, is relatively young and while our findings support its basic premise that using one’s strengths at work is beneficial, our findings call for further exploration of the mechanisms or boundary conditions explaining possible negative outcomes.

In our model we added future-orientedness of one’s job as a resource building on the idea central in sustainable career theory [6] that the time perspective offered by one’s current work might be important for understanding whether a person feels capable of continuing doing their current work in the long run. Our results support the idea that this perspective of future fit is important in explaining perceived work ability as well as perceived meaningfulness of work, thereby underscoring the importance of bringing in a time perspective when researching perceived work ability. Future-oriented jobs are, as expected, associated with perceived meaningfulness of work. This finding underlines that, in order for a job to be perceived as meaningful, both current fit and future fit in terms of long-term prospects and future opportunities need to be present in the current job.

The finding that perceived future fit—not current fit—was the only resource for which the relationship with perceived work ability was moderated by age category, warrants further reflection. In particular, the association was stronger for the mid-career and senior workers compared to the young workers. This finding is consistent with the idea that employees’ perceptions of their work ability are driven by the importance of different foci in life depending on their age category, which is the basic premise of SOC and COR theory ([50,51]). At least in our sample, for the younger workers the future perspective their current job brings them, was less predictive of perceived meaningfulness of their work. One explanation would be that they still see a future full of career opportunities in front of them, making them look further than what might be offered by their current job (cf. an open-ended future time perspective). Of note, there does not seem to be a ‘linear’ relationship between age category and importance of future fit with one’s job. Rather, the mid-career and senior workers did not differ in this regard. The lack of a linear relationship may not be so surprising when considering that the concept of age can take on different meanings in even the same context, such as biological, calendar, psychosocial, organizational, and life-span age [70]). In fact, this is in line with prior findings that chronological age did not have a major influence on future work perceptions over and above psychosocial age [71]. In the context of work ability, these different concepts of age may affect how perceived work ability is affected by future-orientedness of one’s job across the distinguished career stages. The combination of all these different effects may contribute to the non-linear effect we observe. Future research should therefore focus on further examining these differential effects.

The consistent positive and significant relationship of the four resources in our model with perceived meaningfulness of work support notions of positive psychology, which starts from needs-fulfilment being the basis of well-being [68]. Yet, at the same time they suggest, given the lack of a significant relationship between meaningfulness of one’s work and perceived work ability, that more is needed than meaningful work alone for workers to enhance their beliefs about being able to continue doing their work over a longer time period. As such, our findings support the idea of sustainable careers theory that a systemic or multiple-stakeholder perspective is needed to understand why and how employees might be willing and able to continue working, especially in the current labor market where retirement ages are increasing.

Finally, our findings contribute to the relatively new but growing research field of sustainable careers. In their conceptual paper on sustainable careers, De Vos and colleagues [6] formulated several suggestions for empirical research to examine their process model of sustainable careers. Our study responds to their call by studying perceived work ability as an indicator of a sustainable career, thereby bringing in the dimension of time (by including the role of both current and future fit with one’s job). Moreover, we addressed the individual dimension of career sustainability through meaningfulness of work, which is proposed to be important to understand contemporary careers given the increased emphasis on agency and self-management [7]. Interestingly, yet contrasting with our expectation, meaningfulness of work and perceived work ability were not related. This highlights the importance of studying career sustainability from different angles instead of approaching it as a holistic concept. In this paper we focused—in line with the Special Issue—on work ability, which is conceived as being an important, yet only just one, indicator of a sustainable career. Therefore, we cannot draw conclusions regarding the importance of meaningfulness for the other two indicators, i.e., happiness and productivity [6]. It will be interesting for future research to further study the role of resources and meaningfulness of one’s work in explaining the three indicators of a sustainable career, how these are interrelated, and how they might jointly develop over time.

### 4.2. Limitations and Suggestions for Future Research

Our study has several key strengths, including a theoretical expansion of the importance of resources for perceived meaningfulness of one’s work and work ability, thereby considering the moderating role of age. However, there are also several limitations. First, we have theorized and studied perceived work ability in terms of employees’ perceptions of being capable to continue doing their current work now and in the long run. We thereby did not explicitly specify what this time perspective entailed. Moreover, we did not distinguish between the ability to continue working in one’s current *job,* one’s current *profession*, or even continue working *in general*. Hence, future research is needed to better understand how resources might impact perceived work ability when considering various time spans. For example, studies could examine whether low perceptions of work ability might lead employees to engage in job or career transitions in view of increasing their work ability and hence the sustainability of their career.

Second, data was collected using an online cross-sectional survey, and hence common-method bias may exist [62]. To overcome this limitation, we did our best to minimize common-method variance while designing the study, for example, by applying short questionnaires as recommended in procedural methods for reducing common-method bias [62]. A third limitation concerns the use of self-ratings. More scholarly work is needed to better understand how this might have influenced our pattern of results. Against this background, self-ratings to assess our variables seem to have been an appropriate choice given that the constructs used in our model are inherently psychological. In this study we used a self-developed measure to assess perceived work ability. Cronbach’s alpha of this scale was relatively low. One possible explanation for this can be the fact that different age groups may attach different importance to these items. This measure is however closely in line with our operational definition of perceived work ability and not problematically low [72]. When interpreting the lack of support for some of the proposed relationships with perceived work ability, this rather low internal consistency should be kept in mind as low reliabilities tend to attenuate associations between variables.

A first avenue for future research we see is to further elaborate on the relationship between meaningfulness of work and perceived work ability. Contrary to our expectations, and even though both variables were correlated, when testing their relationship in a structural model including the four resources as antecedents, their relationship was no longer statistically significant. Given the potential importance of both meaningfulness of one’s work and work ability for sustainable careers, we suggest that future research further unravels the potential relationship between both variables, thereby including underlying mechanisms such as autonomous versus controlled motivation.

Second, we suggest that future research further explores the role of age in understanding the antecedents of perceived work ability, thereby taking a broader conceptualization of age. Sterns and Doverspike [73] proposed five different approaches comprising chronological, organizational, functional, psychosocial, and life-span development to measure age-related changes, due to health, career stage, and family status, among others, across time. Even if individuals are of the same chronological age, they may still differ in terms of these age-related changes. Therefore, a more elaborate conceptualization of age is needed to better understand its impact on the proposed relationships in our study.

Furthermore, some of the effect sizes in our study were relatively small. For instance, autonomy was only weakly related to the perception of meaningfulness of work. This is in contrast to an abundance of models and literature stressing the importance of autonomy in work-related outcomes, such as the Karasek model [74] and self-determination theory [68]. In this sense, the data of this study can be considered to be an outlier, because direct effects of autonomy in previous work tend to be in the small to medium ranges [75], whereas this study suggests a very small effect. There are a few possibilities that might help us explain this finding. It might simply be the case that autonomy is less important as a predictor of meaningfulness of work when taking into account other resources (i.e., needs-supply fit, strengths use, and future-orientedness of the job), although we assume this to be unlikely given the many studies stressing the importance of autonomy. An alternative explanation is that this is due to the interrelatedness of the resources in our model. The fact that we cannot tease apart the causal order is a weakness that is inherent in utilizing a cross-sectional sample. The structural relationships between antecedents and how these relate to outcomes should be further tested using longitudinal designs. As such, this is another call to action to not only employ research designs that can infer causality to investigate the ‘true’ causal order of predictors, but also to design theory with respect to this internal logic.

Also, we highlight that strengths use will have deeply motivational potential, but might sap resources more quickly than they can recover if not provided in the right context. One can imagine that there are environments in which strengths use can drain the energy out of employees, but that the relationship is situation-dependent or that it interacts with an employee’s personal resources Therefore, we invite researchers to focus on potential boundary conditions, such as human capital related traits and motives, in order to gain more insight into when and why this negative relationship occurs and when it does not.

### 4.3. Practical Implications

This study also has practical implications. First, our findings suggest that resources are important to increase workers’ perceptions of meaningfulness of their work and this is equally important for employees across career stages. Hence, when designing jobs, it is an important question to what extent work allows a person to experience autonomy, use their strengths, feel a fit with their personal values, and have a perspective of future fit. These are all psychological and idiosyncratic variables as they will likely differ between employees. Therefore, we advocate a multiple-stakeholder perspective in which meaningfulness of work is realized through dialogue with all stakeholders involved, that is: the individual workers, line managers, HR, peers, and one’s relatives. There are many individual factors that may impact what affects the meaningfulness of work for a particular employee, and this is likely to be affected by their broader life context and career stage.

Second, even though our central outcome variable—perceived work ability—was not significantly explained by meaningfulness of work, the observations regarding the direct associations between antecedents and perceived work ability deserve attention from a practical standpoint. First of all, experiencing a fit between one’s personal values and what the job offers (i.e., needs-supply fit) is important for perceived work ability, no matter what age category a worker belongs to. This calls for a stronger focus in HR- and people management practices on what a job might bring to a person in terms of needs and values fit. Moreover, the future-orientedness of one’s current job appears to be important for perceived work ability and this is especially the case for employees in mid-career and late-career stages, whose future time perspective is less open ended compared to younger employees who typically perceive ample opportunities in the future. Thus, in order to enhance work ability perceptions for those workers who already have built more seniority in their career, ensuring that they anticipate future fit with their job is important. Focusing on the learning value of the current job [76] will can be an important sustainable career management practice in that regard.

Yet, given the negative association between strengths use and work ability, our study also points out that not all practices focused on increasing the fit with one’s job are equally beneficial in terms of enhancing work ability seen from a sustainable career perspective. HR-managers will thus need to walk a tightrope between motivating employees and ensuring work ability across the lifespan.

## 5. Conclusions

To conclude, this study focused on how workers’ perceptions of current and future fit with their job are related to perceived meaningfulness of work and perceived work ability, and how these relationships might differ according to age. We thereby focused on four types of resources which theoretically represent current fit (i.e., autonomy, strengths use, needs-supply fit) and future fit (i.e., future-orientedness of the job) with one’s job. In line with our hypotheses, these four resources were all related to experienced meaningfulness of work, yet the relationships with perceived work ability were more nuanced. Notably, meaningfulness of work did not function as a mediator in the hypothesized model. Moreover, age only moderated the relationship between future-orientedness of the job and perceived work ability. Together, our findings add to the literature by studying work ability from a sustainable career perspective.

## Figures and Tables

**Figure 1 ijerph-16-02572-f001:**
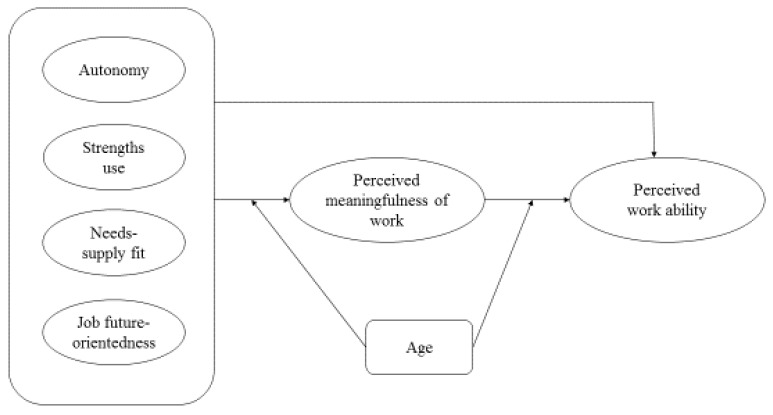
Research model.

**Figure 2 ijerph-16-02572-f002:**
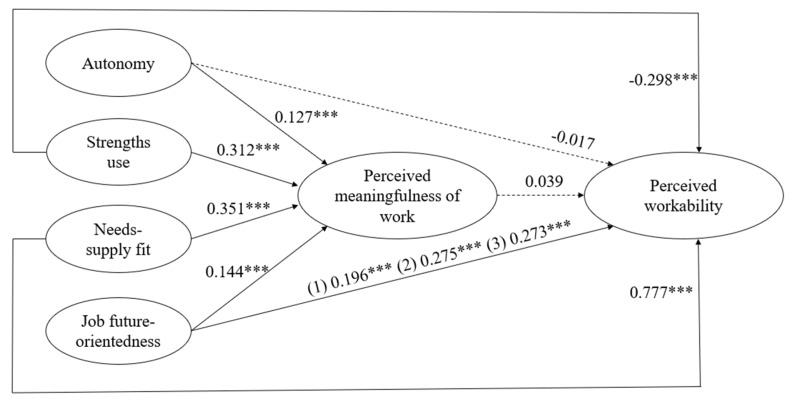
Final model. Note: *** *p* < 0.001. When bèta-weights are the same for the three age categories, only one value is reported (1) = age 20–34; (2) = age 35–49; (3) = age 50+; Fit indices final model: Chi-square (498) = 2777.124, CFI = 0.960, TLI = 0.955, RMSEA = 0.051; Dotted lines represent non-significant relationships.

**Table 1 ijerph-16-02572-t001:** Correlation matrix for the whole sample (structural level): correlations are all significant at *p* < 0.001 level.

	AUT	SU	NSF	FO	MW	WA
Autonomy (AUT)	-	-	-	-	-	-
Strengths Use (SU)	0.721	-	-	-	-	-
Needs-Supply Fit (NSF)	0.671	0.933	-	-	-	-
Future-Orientedness of one’s job (FO)	0.654	0.820	0.826	-	-	-
Meaningfulness of Work (MW)	0.671	0.852	0.846	0.755	-	-
Perceived Work Ability (WA)	0.560	0.752	0.665	0.732	0.713	-

All correlations are significant at *p* < 0.001.

**Table 2 ijerph-16-02572-t002:** Comparison of different models.

	AIC	Χ-Square	DF
Single factor model	254,599.027	8466.270	170
Autonomy	248,831.960	2671.203	156
Needs-Supply Fit			
Strengths Use -> Meaningfulness -> Work Ability (Model A)			
Future-Orientedness			
Autonomy	248,915.481	2760.724	159
Strengths Use -> Needs-Supply Fit -> Meaningfulness -> Work Ability (Model B)			
Future-Orientedness			
Needs-Supply Fit	250,040.437	3887.680	160
Autonomy -> Strengths Use -> Meaningfulness -> Work Ability (Model C)			
-> Future-Orientedness			
Autonomy----------------------------->	248,534.089	2365.332	152
Needs-Supply Fit------------------------->			
Strengths Use -> Meaningfulness -> Work Ability * (Model D)			
Future-Orientedness --------------------------->			

* Final model; Model B is based on the assumption that needs-supply fit is a mediator instead of a separate independent variable. Model C starts from the assumption that autonomy is an ‘enabler’ in the work context and that its effects are mainly expressed through increased strengths use and being able to fit the job better to one’s own needs. Model D is a version of Model 1, but with direct paths added for future-orientedness of one’s job, strengths use, autonomy and needs-supply fit.

**Table 3 ijerph-16-02572-t003:** Test of age interaction.

	Df	Χ-Square	Δ Χ-Square	Δ Df	*p*-Value	Significant after Bonferonni Correction
Model 0: model without constraints	482	2751.3				
Model 1: constrict NSF on MW relation	484	2759.1	7.7798	2	0.02045	
Model 2: constrict SU on MW relation	486	2759.4	0.3207	2	0.85186	
Model 3: constrict FO on MW relation	488	2763.6	4.1380	2	0.12631	
Model 4: constrict AUT on MW relation	490	2764.1	0.5607	2	0.75552	
Model 5: constrict MW on WA relation	492	2769.9	5.8055	2	0.05487	
Model 6: constrict NSF on WA relation	494	2771.6	1.6425	2	0.43988	
Model 7: constrict SU on WA relation	496	2771.8	0.1729	2	0.91718	
Model 8: constrict FO on WA relation	498	2782.8	11.0278	2	0.00403	Yes
Model 9: constrict NSF, but not FO on WA relation +	498	2777.1	5.3645	2	0.06841	

+: compared to Model 7, since Model 8 was not retained due to significant misfit; AUT = Autonomy, NSF = Needs-Supply Fit, SU= Strengths Use, FO = Future Orientedness; MW =Meaningful Work, WA = Work Ability. Bonferonni Correction was set at *p* < 0.0055.

**Table 4 ijerph-16-02572-t004:** Final model standardized effects.

	*β* (Standardized)			
	Between 20 and 34	Between 35 and 49	50+	Significance
Meaningfulness of work				
Autonomy	0.127	.	.	***
Needs-Supply Fit	0.351	.	.	***
Strengths Use	0.312	.	.	***
Future-Orientedness	0.144	.	.	***
Perceived Work Ability				
Autonomy	−0.017	.	.	ns
Needs-Supply Fit	0.777	.	.	***
Strengths Use	−0.298	.	.	***
Future-Orientedness	0.196	0.275	0.272	*** +
Meaningfulness of Work	0.039	.	.	ns

*** *p* < 0.001, +: significance holds for the three age categories; . : Same estimate for other age categories Fit indices final model: Chi-square (498) = 2777.124, CFI = 0.960, TLI = 0.955, RMSEA = 0.051; There is no evidence for mediation, only for moderation, so there can be no moderated mediation.

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
