# Peer review of "A Sustainable Career Perspective of Work Ability: The Importance of Resources across the Lifespan"

_ijerph, 2019, doi:10.3390/ijerph16142572_

Round 1
Reviewer 1 Report
This is an interesting study that relies on a particularly large dataset, despite it's cross-sectional nature. Overall I think the paper constitutes an interesting read for the audience of the journal. Following a few suggestions.
Overall Inputs:
The paper would strongly benefit from a double-check by a native speaker: too long sentences, some sentences have an unusual structure, some words seem out of place and there are some typos.
Introduction and Theory Part:
It seems straightforward that employability is a key factor for sustainable working lives. Be more confident with those statements.
Methods and Results:
Line 292: Instead of stating “Flanders” (I assume it is a city in Belgium) I would speak of “a leading regional newspaper in Belgium”.
Analyses:
Very high intercorrelations between all study variables. I liked that this was nicely balanced by sophisticated analyses and testing for alternative plausible models and repeated factor analyses.
Discussion:
I feel that the discussion part could be substantially improved by:
1) More thorough explanation of the significant negative relationship between strengths use and work ability as well as the other results which were not in line with the hypotheses.
2) Strengthening the discussion section in regards to what other resources might influence work ability and possibly provide ideas for future research to test those assumptions.
Author Response
Dear reviewer
We thank the reviewer for their helpful comments and interest in the study. We believe we addressed the comments raised in an appropriate fashion and we will provide a point by point overview of the comments and how we addressed them.
Comments and Suggestions for Authors
This is an interesting study that relies on a particularly large dataset, despite it's cross-sectional nature. Overall I think the paper constitutes an interesting read for the audience of the journal. Following a few suggestions.
Overall Inputs:
The paper would strongly benefit from a double-check by a native speaker: too long sentences, some sentences have an unusual structure, some words seem out of place and there are some typos.
In response to your concern about writing deficiencies the paper also has been carefully reviewed and writing deficiencies have been addressed. We shortened sentences and applied a general language check.
Introduction and Theory Part:
It seems straightforward that employability is a key factor for sustainable working lives. Be more confident with those statements.
Dear Reviewer, we have now stressed the importance of employability and have linked it to the indicator of productivity in the following text excerpt (see line 132):
“Productivity means strong performance in one's current job as well as a guaranteeing a high employability or career potential towards the future [18].”
Methods and Results:
Line 292: Instead of stating “Flanders” (I assume it is a city in Belgium) I would speak of “a leading regional newspaper in Belgium”.
Flanders is the Dutch speaking region in Belgium, in order to clarify this, we changed the sentence to (l365-366):
“Data were collected in collaboration with a leading newspaper in Flanders which is the Dutch speaking region in Belgium.”
Analyses:
Very high intercorrelations between all study variables. I liked that this was nicely balanced by sophisticated analyses and testing for alternative plausible models and repeated factor analyses.
Thank you, we indeed felt this was necessary given the combination of cross-sectional design mixed with high intercorrelations.
Discussion:
I feel that the discussion part could be substantially improved by:
1) More thorough explanation of the significant negative relationship between strengths use and work ability as well as the other results which were not in line with the hypotheses.
We recognize the issue and per your suggestion we have extended the discussion to accommodate this. Concretely we added a possible theoretical explanations for these findings starting from l597 to 620, the excerpt now reads as follows:
“The third indicator of current fit with one’s job, strengths use, also showed surprising results. Although it related positively to meaningfulness of work as hypothesized, contrary to our expectations, it related negatively to perceived work ability. Initially, we were surprised with this finding given prior evidence for strengths use as a predictor of well-being [66] However, there is some empirical evidence suggesting that in order for work to be deeply meaningful one also needs to ‘suffer’ for their craft [67]. In line with this argumentation, prior studies found that challenging job demands can work both as a motivator and a stressor [68]. Consequently, one can speculate that strengths use, besides being an attractive resource, may also instill these potentially harmful aspects and inspire people to work ‘to the bone’. Thus, even though a high level of strengths use is likely to provide a sense of meaningfulness in one’s work, it can also have a potential dark side of undermining work ability when people are too highly involved in their job. Furthermore, the sustainable career paradigm may also provide a further explanation. From this perspective, personal investment in one’s current job might lead to depletion of resources thereby lowering the sustainability of one’s career over time [6]. Reduced health – operationalized in this study in terms of perceived work ability – might be an important indicator of this phenomenon. A related explanation can be found in research on workaholism, from which we know that the mechanism of controlled motivation might explain the negative impact of ‘working too hard’ on employee outcomes, compared with the positive impact of engagement[69,70]. The research on strengths use, stemming from the domain of positive psychology, is relatively young and while our findings support its basic premise that using one’s strengths at work is beneficial, our findings call for further exploration of the mechanisms or boundary conditions explaining possible negative outcomes.”
2) Strengthening the discussion section in regards to what other resources might influence work ability and possibly provide ideas for future research to test those assumptions.
We agree with this point and see several avenues for future research. We refer to the paper l686-753 for our adjustments to the suggestions for future research. In particular, we suggest that personal resources may interact with work-related resources in explaining work ability.
Reviewer 2 Report
The research investigates the determinants of Work Ability (WA) in a cross-sectional design adopting the SEM framework. The topic is of interest for public health, and the theoretical approach (Sustainable Career Perspective) is promising.
I hope my comments are helpful for the authors in developing their ideas.
Introduction
Workability is a concept measured with many different instruments from several theoretical backgrounds (e.g., Lederer et al., 2014). It is, therefore, a very complex field of research, and all the researchers adopting the concept of WA should carefully define what they mean when they refer to it. The present study starts with a clear and well-known definition of WA.
Unfortunately, WA is then extended in its conceptual meaning so that the reader is lost wondering between the first definition of WA and the second new version of WA, sometimes called sustainable WA.
Given that the authors adopt the Sustainable career perspective, WA is extended in its time perspective toward the far future (the whole working life). This extension is a cornerstone to the comprehension of the study, but it is not well explained and substantiated in the introduction and paragraph 1.1. The critical points are:
A) The authors should justify how WA is consistent with the long-term perspective of the Sustainable career perspective. If it is true, as stated by the authors that “workability can be characterized as a worker’s general feelings or perceptions regarding their capability to continue doing their current work.” It is also true that “to continue doing their current work” refers to the present continuous or the near future, and surely the current job, not the whole career. WA is determined by the current context and working conditions (as you wrote “the work environment and community as well as the actual contents and organization of their work”). A good workability indicates a good balance in the current job, but it is not clear what information it gives about the next job. If not explained, WA seems related to the concept of a sustainable job more than a sustainable career. There is just a reference [9] connecting WA with retirement. However, (1) the reference is not precise as [9] is an editorial citing an article relating WA with retirement (Fadyl, et al. 2010) and (2) this article presents some methodological limits when studying retirement. If the author like, I suggest using this reference: Jääskeläinen et al. (2016) that seems stronger to me. Even with one correct reference to an article relating WA to retirement, I suspect this is not enough to demonstrate that WA, measured once, is a clear index of a sustainable career (instead of a sustainable job).
B) WA is presented as an indicator of a sustainable career, but this should be better justified. The reference to the model of sustainable careers should be used to justify the use of WA and at the same time to highlight the partiality of this concept and its limitations. That is, WA is one way to measure at the individual level just one (health) of the three elements that suggest a sustainable career (health, happiness, and productivity). This limit should be clearly explained in the introduction since the authors say to adopt the sustainable career perspective, but there is just a quick reference to this in the discussion. On the opposite, a clear and accurate definition of WA (including its limits) is not present.
C) WA seems to assume a too great theoretical role in relation to sustainable career. On Paragraph 1.1, that define the concept, WA assumes the characteristics of the bigger concept Sustainable career of which WA is just a partial indicator. For example, “work ability refers to the individual’s capacity to optimally balance personal resources and work demands, across the career life-span, and to continue working, while at the same time protecting one’s health, happiness, and productivity” is more likely the description of what characterizes a sustainable career. Moreover, the reference [6] used next to the definition of WA does not provide any definition of it. Instead [6] suggests that indicators of health, happiness, and productivity should be used together (e.g., work ability, engagement, and employability) “because they are interrelated and together they characterize the sustainability of careers”.
With this saying, I do not mean that there is no value in studying work ability, but it is necessary to give the correct theoretical framework related with the concept used, also to allow a better interpretation of results.
D) In case the authors are proposing a new form of WA this should be better presented, and differences with the current versions of WA should be highlighted.
Given that the concept of sustainable WA is not clearly defined, the subsequent hypothesized relations of WA with the predictors remain quite complex to understand.
l.270-271 It is not clear how from the SOC theory derives that a meaningful job became more relevant for identity among elderly than young workers
l.277 As correctly reported by authors SST states that people prioritize meaningfulness of interactions because their future time perspective is starting to get limited. Then, I do not understand why the authors wrote that the elderly prefer affective rewards because these strengthen their identity.
l.278 It is not explained why to behave more authentically should increase WA
Methods
l.313 The citation for the autonomy measure is missing.
l.325 It is not clear how WA is operationalized. I was not able to find any item related to WA in the study cited as a reference for the three items [56]. Because WA is the dependent variable, and the measure is new, it is necessary to give more information about it. For example, it would be useful to list all three items used. Moreover, α is low, and an explanation about this would help.
l.335 It is useful to explicit in the methods the criteria used to establish metric invariance.
Results
I have some problem in calculating the degree of freedom of the models. Specifically, considering the initial measurement model the df reported is 155 but with 21 observed variables and 6 latent variables I expected a df= [21(21+1)/2] – (21 loadings + 21 error variances + 15 correlations) =174. Are the df correct?
I also noticed that df of the structural models (table 2) are higher than the one of the measurement model chosen (df=152). This is not possible unless they are not full SEMs but path analyses. Is it possible to clarify this point?
l.366 The second DeltaCFI is DeltaTLI.
l.390 and table2 There is inconsistency about how the models are called, with letters or numbers
Discussion
l.442-458 This part can be removed to reduce the length of the article.
References from 64 are not present in the reference list.
The limits section have to be improved.
l.574 “in SEM-modeling” should be removed as the problem seems to be SEM more than the research design.
l.576 The sentence did not conclude.
Lederer, V., Loisel, P., Rivard, M., & Champagne, F. (2014). Exploring the diversity of conceptualizations of work (dis)ability: A scoping review of published definitions. Journal of Occupational Rehabilitation, 24(2), 242–267. http://doi.org/10.1007/s10926-013-9459-4
Jääskeläinen, A., Kausto, J., Seitsamo, J., Ojajärvi, A., Nygård, C. H., Arjas, E., & Leino-Arjas, P. (2016). Work ability index and perceived work ability as predictors of disability pension: A prospective study among Finnish municipal employees. Scandinavian Journal of Work, Environment and Health, 42(6), 490–499. https://doi.org/10.5271/sjweh.3598
Author Response
Dear reviewer
Thank you for your detailed reading of our paper and the interest you show about the relevance of our research topic. Following your review, we have addressed a few major issues. First, we made substantial changes to the introduction, in order to get a more coherent story and especially focused on our conceptualization of workability as a concept. Secondly, we have worked on our contributions and the boundaries of our study. Thirdly, following your suggestion, we have made improvements to the method section. We thank you for the very constructive and helpful review letter and for your insightful feedback. We will now respond point-by-point how we have dealt with your comments.
Introduction
Workability is a concept measured with many different instruments from several theoretical backgrounds (e.g., Lederer et al., 2014). It is, therefore, a very complex field of research, and all the researchers adopting the concept of WA should carefully define what they mean when they refer to it.
We agree that our current framing and conceptualization of WA was too complex and not always consistent. We are thankful for your insightful comments in that regard. Following your feedback we have removed some pieces of text in which we elaborated too far on WA from a sustainable career perspective. Following your feedback on our conceptualization of WA we have reworked parts of the introduction section to make it more clear to the reader that we are focusing on perceived WA, referring to a person’s general perceptions regarding their ability to continue working. Therefore we have, amongst others, removed L53-57 (original version of the paper) as this addition might have brought more confusion than clarity to our line of reasoning. We have removed reference to “sustainable WA” as it was not our intention to introduce this as a specific form of WA.
The present study starts with a clear and well-known definition of WA. Unfortunately, WA is then extended in its conceptual meaning so that the reader is lost wondering between the first definition of WA and the second new version of WA, sometimes called sustainable WA.
Given that the authors adopt the Sustainable career perspective, WA is extended in its time perspective toward the far future (the whole working life). This extension is a cornerstone to the comprehension of the study, but it is not well explained and substantiated in the introduction and paragraph 1.1. The critical points are:
A) The authors should justify how WA is consistent with the long-term perspective of the Sustainable career perspective. If it is true, as stated by the authors that “workability can be characterized as a worker’s general feelings or perceptions regarding their capability to continue doing their current work.” It is also true that “to continue doing their current work” refers to the present continuous or the near future, and surely the current job, not the whole career. WA is determined by the current context and working conditions (as you wrote “the work environment and community as well as the actual contents and organization of their work”). A good workability indicates a good balance in the current job, but it is not clear what information it gives about the next job. If not explained, WA seems related to the concept of a sustainable job more than a sustainable career. There is just a reference [9] connecting WA with retirement. However, (1) the reference is not precise as [9] is an editorial citing an article relating WA with retirement (Fadyl, et al. 2010) and (2) this article presents some methodological limits when studying retirement. If the author like, I suggest using this reference: Jääskeläinen et al. (2016) that seems stronger to me. Even with one correct reference to an article relating WA to retirement, I suspect this is not enough to demonstrate that WA, measured once, is a clear index of a sustainable career (instead of a sustainable job).
You are correct that this future time perspective does not distinguish between the current job and possible other jobs in the future. Our definition and operationalization follows from the framing of WA from a career perspective. Time is an essential component of any career, and careers can be studied by looking at how career sequences evolve over a longer time period, or by studying how individuals perceive their career by looking backward or forward. In our paper we take this forward looking perspective yet without specifying this time frame. On the one hand, this is consistent with career research and how the element of time is brought in (e.g., for future time perspective “the future” is also not defined and can be interpreted idiosyncratically by the respondent). On the other hand this is indeed a shortcoming of our current operationalization and we have therefore added this in our discussion section when discussing the limitations of our study. We explicitly want to thank you for giving more relevant references, we included these in our paper.
B) WA is presented as an indicator of a sustainable career, but this should be better justified. The reference to the model of sustainable careers should be used to justify the use of WA and at the same time to highlight the partiality of this concept and its limitations. That is, WA is one way to measure at the individual level just one (health) of the three elements that suggest a sustainable career (health, happiness, and productivity). This limit should be clearly explained in the introduction since the authors say to adopt the sustainable career perspective, but there is just a quick reference to this in the discussion. On the opposite, a clear and accurate definition of WA (including its limits) is not present.
Thank you once again for your constructive and helpful comments to provide a clearer and more consistent conceptualization of WA in our manuscript. We now introduce and consistently refer to our definition of WA as “a worker’s general feelings or perceptions regarding their capability to continue doing their current work.” This is also how it is measured in our study. We have reworked the first part of the paper to make it clear why and how we approach WA from a sustainable career perspective, rather than aiming to introduce a new conceptualization of WA. Indeed, we see many possible linkages between the literature on WA and the (sustainable) careers literature and believe this integration forms the major contribution of our work. We have provided some more convincing arguments for this in the first section, by also better explaining and framing WA as being one indicator of, yet not being the same as, a sustainable career, and what sustainable career theory can bring to further our understanding of WA. We also elaborate further upon this in the discussion section.
Line 55 of the revised paper now states as follows:
“In particular, in this paper we study perceived work ability, referring to it as a worker’s general feelings or perceptions regarding their capability to continue doing their current work towards the future. Work ability is a holistic concept that refers to peoples’ ability to do their work in a healthy and productive way given the balance between a person’s resources – including their health and functional abilities, education and competence, and values and attitudes – and their work demands [9–11]. As such, at its core, ‘perceived work ability’ refers to employees’ perceived ability to work’.”
C) WA seems to assume a too great theoretical role in relation to sustainable career. On Paragraph 1.1, that define the concept, WA assumes the characteristics of the bigger concept Sustainable career of which WA is just a partial indicator. For example, “work ability refers to the individual’s capacity to optimally balance personal resources and work demands, across the career life-span, and to continue working, while at the same time protecting one’s health, happiness, and productivity” is more likely the description of what characterizes a sustainable career.
Dear Reviewer, we have changed this section in line with your comment and the text excerpt now reads as follows (L119-173 of the revised paper):
“The idea that careers reflect the continued employment of individuals in jobs that facilitate their personal development over time has been the underlying ideology of career research for a long time [16]. Analogously, the notion of sustainable careers addition approaches the career from a dynamic and systemic perspective, arguing that multiple stakeholders play a key role, such as one’s family, peers, supervisor, and employer [6]. As such, the sustainable career paradigm considers how a person can foster person-career fit over time by generating new resources through one’s work rather than depleting them [6,7]. Health, happiness and productivity [17] are considered as the three indicators of a sustainable career. Health encompasses both physical and mental health, and refers to the dynamic fit of the career with one's mental and physical capacities. Happiness concerns the dynamic fit of the career with one's values, career goals, and needs. Productivity means strong performance in one's current job as well as a guaranteeing a high employability or career potential towards the future [18].
Seen from a sustainable career perspective, perceived work ability refers to the individual’s general feelings or perceptions to continue performing their current work. More specifically, it expresses how well the individual resources meet the requirements of the job ([19], p. 393). Perceived work ability implies the anticipated experience of balance between personal resources and work demands across the career, and has found to be a strong predictor of early retirement from the labor market [19–21]. We argue that perceived work ability is an important element of a person’s long-term health, and thereby forms an indication of someone’s career sustainability [6].
To achieve high levels of work ability, it is important that an employee is mindful about what matters to them [12], and creates opportunities for a meaningful existence [22,23]. As such, perceived work ability has two important components: a developmental and an individual component [6]. First, the developmental component underlines that the employee builds upon and expands their resources over a longer period of time, preferably across the entire career. This puts an emphasis on how well one’s work protects and enhances one’s current as well as one’s future perceptions of work ability, thus incorporating the capacity to flourish both in the here-and-now and in the future.
Second, the individual component of perceived work ability underlines that work should lead to a personally meaningful existence, and should be understood from a fundamentally individualistic perspective [24], as careers form a complex mosaic of objective experiences and subjective evaluations [6]. This complexity has a strong impact on the meaningfulness of one’s work, and explains its highly idiosyncratic character (ibid.).
Consistent with these arguments, in this study we focus on the association between resources and perceived work ability, thereby incorporating not only a person’s perceptions of one’s current but also of one’s future fit with the job. Moreover, in this relationship, we expect perceived meaningfulness to play a critical role as a mediator, and we posit that age might moderate the pattern of relationships.”
Moreover, the reference [6] used next to the definition of WA does not provide any definition of it. Instead [6] suggests that indicators of health, happiness, and productivity should be used together (e.g., work ability, engagement, and employability) “because they are interrelated and together they characterize the sustainability of careers”. With this saying, I do not mean that there is no value in studying work ability, but it is necessary to give the correct theoretical framework related with the concept used, also to allow a better interpretation of results.
In response to your concerns we now provide a stronger and more consistent framing of WA from a sustainable careers perspective, rather than suggesting to introduce a new form of WA.
D) In case the authors are proposing a new form of WA this should be better presented, and differences with the current versions of WA should be highlighted. Given that the concept of sustainable WA is not clearly defined, the subsequent hypothesized relations of WA with the predictors remain quite complex to understand.
We apologize for the confusion that we may have caused by using the term ‘sustainable WA’ in our manuscript. To clarify, we did not mean to introduce a new WA concept. Rather, we approach WA, as it is currently conceptualized, from a sustainable career perspective. We hope that by the substantial changes we have made to the first section of the paper this has become more clear and we thank you for your helpful suggestions in that regard.
l.270-271 It is not clear how from the SOC theory derives that a meaningful job became more relevant for identity among elderly than young workers
The reference to ‘identity’ might have been confusing here. What we meant to say is that the experienced meaningfulness of the current job will be more important for older employees as they are focused more, from a SOC perspective, on retaining what they currently have in their job than on finding new sources of meaningfulness or fulfillment in other jobs. We addressed this concern in the paper (see L333-341 of the revised paper):
“Correspondingly, Freund [53] found a shift in regulatory focus from being aimed at promotion for younger individuals to focusing on maintenance and prevention in later life. In the context of our study, we argue that meaningfulness in one’s job will be more important for older employees as they are relatively more focused on maintaining what they currently have and preventing losses, compared with younger employees who are more focused on striving for future opportunities and growth in one’s current or in other jobs.”
l.277 As correctly reported by authors SST states that people prioritize meaningfulness of interactions because their future time perspective is starting to get limited. Then, I do not understand why the authors wrote that the elderly prefer affective rewards because these strengthen their identity.
We have removed this part of the sentence as it was indeed not representing the core of our argument (see track changes L341 of the revised paper).
l.278 It is not explained why to behave more authentically should increase WA
We removed this in the paper, as it has no added value to our core argument (see track changes L347 of the revised paper)..
l.313 The citation for the autonomy measure is missing.
Reference has been added, thank you for noticing!
l.325 It is not clear how WA is operationalized. I was not able to find any item related to WA in the study cited as a reference for the three items [56]. Because WA is the dependent variable, and the measure is new, it is necessary to give more information about it. For example, it would be useful to list all three items used. Moreover, α is low, and an explanation about this would help.
To capture work ability from a sustainable career perspective, we wanted to measure the degree to which respondents felt capable of continuing doing their current work. To do so, we used the following three items: “I don’t see myself continuing to work in my current job for much longer” (reversed scoring), “I feel able to continue working my current job until I retire” and “A higher retirement age is not a problem for me personally”. Cronbach’s alpha was 0.61, which is indeed on the low side. We do believe that this can be partially be explained due to the fact that different age groups attach a differential importance to these items as will be explained later. This measure is however closely in line with our operational definition of work ability and not problematically low. Furthermore, it can be seen as a conservative test, since low reliabilities tend to attenuate associations between variables.
l.335 It is useful to explicit in the methods the criteria used to establish metric invariance.
Indeed. We have added the drop in global fit indices as our criteria for establishing metric invariance. We added the following to the text to clarify (l418-420):
“Metric invariance was established by comparing the change in global fit indices. One could use a Chi-square-test, but this might lead to an oversensitive test given our large sample size.”
Results
I have some problem in calculating the degree of freedom of the models. Specifically, considering the initial measurement model the df reported is 155 but with 21 observed variables and 6 latent variables I expected a df= [21(21+1)/2] – (21 loadings + 21 error variances + 15 correlations) =174. Are the df correct?
We thank the reviewer for being so thorough with recalculations, our measures included 20 observed constructs total, instead of 21 (due to a typo in the number of items in meaningful work). We addressed this issue in the method section. As such, the degrees of freedom are correct in the paper (using the formula: [20(20+1)/2 – (20 + 20 + 15)).
I also noticed that df of the structural models (table 2) are higher than the one of the measurement model chosen (df=152). This is not possible unless they are not full SEMs but path analyses. Is it possible to clarify this point?
To our understanding, CFA are standardly ‘saturated’ at the latent level, meaning that all degrees of freedom are ‘taken away’ at the latent level (all latent variables are correlated with each other). For a structural model one wants a more parsimonious description of the data than a CFA provides. Simpler models tend to not be saturated at the latent level, resulting in a higher number of degrees of freedom than a CFA. As such, it is normal that there are more degrees of freedom for the structural model than the CFA model. Hopefully this explanation addresses your concern.
l.366 The second DeltaCFI is DeltaTLI.
We corrected this, thank you.
l.390 and table2 There is inconsistency about how the models are called, with letters or numbers
This was indeed the case, now it is consistently labeled in the table. Thank you for notifying this and we apologize for the confusion this might have caused.
Discussion
l.442-458 This part can be removed to reduce the length of the article.
Per your suggestion, we have removed these lines in order to reduce length.
References from 64 are not present in the reference list.
We apologize for this, something must have gone wrong when uploading our paper as they were present in our original Word document. We have added them to the paper.
The limits section have to be improved.
We added several improvements to this section. It can be read in the paper from lines 686-753. We thereby address both conceptual and methodological limitations and also address the low Cronbach alpha score for our perceived work ability measure. The revised section reads as follows:
“Our study has several key strengths, including a theoretical expansion of the importance of resources for perceived meaningfulness of one’s work and work ability, thereby considering the moderating role of age. However, there are also several limitations. First, we have theorized and studied perceived work ability in terms of employees’ perceptions of being capable to continue doing their current work now and in the long run. We thereby did not explicitly specify what this time perspective entailed. Moreover, we did not distinguish between the ability to continue working in one’s current job, one’s current profession, or even continue working in general. Hence, future research is needed to better understand how resources might impact perceived work ability when considering various time spans. For example, studies could examine whether low perceptions of work ability might lead employees to engage in job or career transitions in view of increasing their work ability and hence the sustainability of their career.
Second, data was collected using an online cross-sectional survey, and hence common-method bias may exist [63]. To overcome this limitation, we did our best to minimize common-method variance while designing the study, for example, by applying short questionnaires as recommended in procedural methods for reducing common-method bias [63]. A third limitation concerns the use of self-ratings. More scholarly work is needed to better understand how this might have influenced our pattern of results. Against this background, self-ratings to assess our variables seem to have been an appropriate choice given that the constructs used in our model are inherently psychological. In this study we used a self-developed measure to assess perceived work ability. Cronbach alpha of this scale was relatively low. One possible explanation for this can be the fact that different age groups may attach different importance to these items. This measure is however closely in line with our operational definition of perceived work ability and not problematically low [73]. When interpreting the lack of support for some of the proposed relationships with perceived work ability, this rather low internal consistency should be kept in mind as low reliabilities tend to attenuate associations between variables.
A first avenue for future research we see is to further elaborate on the relationship between meaningfulness of work and perceived work ability. Contrary to our expectations, and even though both variables were correlated, when testing their relationship in a structural model including the four resources as antecedents, their relationship was no longer statistically significant. Given the potential importance of both meaningfulness of one’s work and work ability for sustainable careers, we suggest that future research further unravels the potential relationship between both variables, thereby including underlying mechanisms such as autonomous versus controlled motivation.
Second, we suggest that future research further explores the role of age in understanding the antecedents of perceived work ability, thereby taking a broader conceptualization of age. Sterns and Doverspike [74] proposed five different approaches comprising chronological, organizational, functional, psychosocial, and life-span development to measure age-related changes, due to health, career stage, and family status, among others, across time. Even if individuals are of the same chronological age, they may still differ in terms of these age-related changes. Therefore, a more elaborate conceptualization of age is needed to better understand its impact on the proposed relationships in our study.
Furthermore, some of the effect sizes in our study were relatively small. For instance, autonomy was only weakly related to the perception of meaningfulness of work. This is in contrast to an abundance of models and literature stressing the importance of autonomy in work-related outcomes, such as the Karasek model [75] and self-determination theory [69]. In this sense, the data of this study can be considered to be an outlier, because direct effects of autonomy in previous work tend to be in the small to medium ranges [76], whereas this study suggests a very small effect. There are a few possibilities that might help us explain this outcome. It might simply be the case that autonomy is less important as a predictor of meaningfulness of work when taking into account other resources (i.e., needs-supply fit, strengths use, and future-orientedness of the job), although we assume this to be unlikely given the many studies stressing the importance of autonomy. An alternative explanation is that this is due to the interrelatedness of the resources in our model. The fact that we cannot tear apart the causal order is a weakness that is inherent in utilizing a cross-sectional sample. The structural relationships between antecedents and how these relate to outcomes should be further tested using longitudinal designs. As such, this is another call to action to not only employ research designs that can infer causality to investigate the ‘true’ causal order of predictors, but also to design theory with respect to this internal logic.
Also, we highlight that strengths use will have deeply motivational potential, but might sap resources more quickly than they can recover if not provided in the right context. One can imagine that there are environments in which strengths use can drain the energy out of employees, but that the relationship is situation-dependent or that it interacts with an employee's personal resources. Therefore, we invite researchers to focus on potential boundary conditions, such as human capital related traits and motives, in order to gain more insight into when and why this negative relationship occurs and when it does not.”
l.574 “in SEM-modeling” should be removed as the problem seems to be SEM more than the research design.
Indeed, we recognize that it is a weakness inherent to the design and corrected the sentence.
l.576 The sentence did not conclude.
Thank you for notifying. The sentence now reads as following to accommodate this issue:
“The structural relationships between antecedents and how these relate to outcomes should be further tested using longitudinal designs”
Round 2
Reviewer 2 Report
Dear authors,
I enjoyed the new version of the article. You answered all my concerns, thanks for your efforts.
I have a few minor requests:
Please check all the manuscript for typo such as in line 623
L419 What change in the global fit indices was considered acceptable?
L465-469 At the beginning it is said that the item has higher loading in two of the three age categories than it is said that the item has a lower loading in two of the categories.
All the best,
Author Response
Dear reviewer
Please see the attachment for our response to your comments.
Kind regards
David
